# Investigating the Life Expectancy at Birth of Companion Dogs in Portugal Using Official National Registry Data

**DOI:** 10.3390/ani14152141

**Published:** 2024-07-23

**Authors:** Helena Geraz, Katia Pinello, Denisa Mendonça, Milton Severo, João Niza-Ribeiro

**Affiliations:** 1Instituto de Ciências Biomédicas Abel Salazar (ICBAS), Universidade do Porto, Rua de Jorge Viterbo Ferreira 228, 4050-313 Porto, Portugal; kcpinello@icbas.up.pt (K.P.); dvmendon@icbas.up.pt (D.M.); jjribeiro@icbas.up.pt (J.N.-R.); 2Departamento de Ciências da Saúde Pública e Forenses e Educação Médica, Faculdade de Medicina da Universidade do Porto, Alameda Professor Hernâni Monteiro, 4200-319 Porto, Portugal; 3EPIUnit-Instituto de Saúde Pública, Universidade do Porto, Rua das Taipas 135, 4050-600 Porto, Portugal; msevero@icbas.up.pt; 4Vet-OncoNet, Departamento de Estudo de Populações, Instituto de Ciências Biomédicas Abel Salazar (ICBAS), Universidade do Porto, Rua de Jorge Viterbo Ferreira 228, 4050-313 Porto, Portugal

**Keywords:** canine longevity, life expectancy, companion dogs, breed size, skull shape

## Abstract

**Simple Summary:**

This study considers several factors that may impact canine lifespan, including breed, sex, size, and skull shape. The results reveal that Portuguese dogs typically live for a mean of 8.91 years, with females living slightly longer than male dogs. Additionally, smaller breeds tend to have a longer lifespan compared to larger ones, while brachycephalic (short-nosed) breeds exhibit the lowest life expectancy. These findings enhance the understanding of the factors influencing canine longevity and aid in developing strategies to improve the health and lifespan of companion dogs.

**Abstract:**

This study aimed to provide a comprehensive picture of the life expectancy of dogs in Portugal, focusing on the impact of diverse factors including breed, sex, size, and skull shape. The final dataset, gathering data from the national registry database, consisted of 278,116 dogs with confirmed deaths. The mean lifespan at birth for all the dogs was around 8.91 years, with the female dogs tended to have a similar lifespan to male dogs. The analysis of life expectancy at birth for the 20 most common non-Portuguese breeds and 10 Portuguese breeds revealed that Yorkshire Terriers had the highest life expectancy (10.89 years) and French Bulldogs the lowest (6.27 years). Size and cephalic index were found to be influential factors, with large brachycephalic breeds exhibiting shorter life expectancies and smaller, mesocephalic breeds experiencing longer lifespans. Additionally, the cephalic index had a more substantial impact on life expectancy compared to body size. These findings enhance the understanding of the factors influencing canine longevity and aid in developing strategies to improve the health and lifespan of companion dogs.

## 1. Introduction

Dogs, as companion animals, serve as models for understanding aging, morbidity, and longevity, shedding light on the biological and environmental factors that impact these processes [1]. Research on the longevity of mixed-breed and purebred dogs has yielded mixed results, with most studies suggesting that mixed-breed dogs live longer than purebred dogs [2,3,4,5,6]. However, a more recent study [7] has cast some uncertainty on this notion, revealing that almost half of purebred dogs (47.1%) actually had a longer median survival estimate than crossbreds. Moreover, only 25.8% of the purebred dogs had a shorter lifespan and 27.1% did not show any significant difference in lifespan when compared to mixed-breed dogs.

Studies have consistently shown a correlation between body size and lifespan in dogs, with smaller breeds displaying a tendency to live longer than larger breeds [4,7,8,9,10]. Furthermore, a recent study using data from the Dog Aging Project [11] found that dog size, along with age, is a significant predictor of disease risk, with larger dogs being more susceptible to various diseases, including skin problems, orthopedic issues, gastrointestinal problems, ear/nose/throat issues, cancer, neurological disorders, endocrine disorders, and infectious diseases.

Most dog breeds tend to have similar lifespan patterns, with a lower mortality rate during young adulthood that gradually increases, peaking in their senior years [4,8,12,13]. Additionally, studies have highlighted significant similarities in the age-related disease risk between humans and dogs, with causes of death such as neoplastic, congenital, and metabolic factors following comparable age trajectories in both species [1].

There is no universally adopted standard for how dog mortality data should be collected, recorded, and reported to allow for accurate comparisons across studies and populations. Two commonly used methods in survival analysis are the Kaplan–Meier analysis, Cox proportional hazards regression [14], and the Chiang method using life tables [15].

The Cox regression and Kaplan–Meier methods are often used in longitudinal studies focusing on time-to-event outcomes, allowing for the inclusion of multiple covariates. These methods typically deal with both censored and uncensored data. Censoring occurs when the event is not observed. Using these methods, the median survival time can be compared across groups even if not all individuals have a recorded date of death [16,17].

Life tables are commonly used in demography, epidemiology, and public health to estimate survival rates, mortality rates, and life expectancy [18]. There are two primary types of life tables—cohort life tables and current life tables. Each offers a unique perspective on mortality rates and life expectancy, allowing for the analysis of different aspects of population dynamics. A cohort life table, also known as a generation life table, tracks the mortality experience of a specific group over their entire lifetime. It provides a detailed picture of how mortality rates change over time for a particular generation, considering both observed and projected improvements in mortality [19]. On the other hand, a current or period life table offers a hypothetical representation of a group’s mortality experience if they were exposed to the mortality rates of a specific time period throughout their whole lifespan. It is based on the death rates prevailing in that year, regardless of the age of the individuals experiencing those deaths. Period life tables can be created using more recent data, allowing for a more up-to-date picture of mortality trends [20].

Creating a comprehensive life table represents a unique challenge, primarily due to the need for a vast dataset that captures the diverse breed structures and health factors present across different canine populations. The task’s complexity is further exacerbated by variations in breeding practices, environmental influences, genetic and health-related factors [6,16,21,22], and access to veterinary care, which can vary significantly between geographical regions [23,24].

Despite these challenges, researchers in several countries have made valuable contributions to this field. Studies conducted in Japan [2,25,26], the United States [3], and the United Kingdom [27] have developed life tables to calculate life expectancy for dogs, shedding light on the longevity and survival patterns within specific populations. To date, no published research has specifically addressed this topic within the Portuguese context, presenting an opportunity for further investigation and contribution to the body of knowledge on canine longevity and survival in different cultural and geographical settings. The objective of this study was to use data from the official national registry dog population database to analyze the life expectancy at birth of the Portuguese companion dog population. For the first time, period life tables were created for the 20 most common non-Portuguese and 10 Portuguese dog breeds. This study also aimed to identify significant variations in life expectancy at birth, considering factors such as sex, breed, body size, and skull shape.

## 2. Materials and Methods

### 2.1. Study Population and Data Extraction

This study population encompassed all dogs listed in the official recognized national registry, the Companion Animal Information System database (SIAC) [28].

In Portugal, according to Decree No. 312/2003 and Decree No. 315/2009, microchipping is mandatory for all companion dogs born after 1 July 2008. All microchipped dogs are registered in the SIAC database, creating a comprehensive national registry for all dogs in the country.

To protect privacy, all records were anonymized by encrypting each animal’s identification number. To maintain accurate and up-to-date information, the registry should be updated in the event of a dog’s disappearance, relocation, transfer of ownership, or death.

The data extracted for this study included demographic information (breed, sex, date of birth, neutered status, and date of death) and geographic location (parish, county, and district of owner’s residence). Owner-reported parameters, including the date of birth and pedigree status, are provided at the time of the dog’s registration in the database. The lifespan was calculated based on the period from the reported date of birth to the date of death.

### 2.2. Data Cleaning and Statistical Analyses

To guarantee the precision of the data, the database underwent a cleaning process involving the removal of specific records meeting the following criteria: (a) deceased before 1 January 2013 or after 31 December 2022, (b) with a negative lifespan, (c) lacking essential birth or mortality information, and (d) missing breed or sex details. Since microchipping became mandatory after 2008, we focused on records from 2013 onwards to take advantage of the increased data volume and reliability. This strategic timeframe ensured data consistency and completeness, reflecting the era of mandatory microchipping. Additionally, a ten-year period provided a longer-term perspective, smoothing out short-term fluctuations and yielding more accurate estimates of mortality and survival rates and life expectancy.

The official recording of animal deaths in the database was considered inadequate, leading to the potential existence of “phantom” entries where animals were inaccurately labeled as “alive” or implausibly old. To address this uncertainty caused by delayed death notifications, which often overestimated lifespans and delayed birth notifications, which tended to underestimate lifespans in certain breeds, we adopted a two-stage approach. First, guided by prior assumptions, we identified three possible groups, namely delayed death notifications (“phantom” entries), delayed birth notifications, and not delayed death and birth notifications (“standard”). Next, we employed outlier modeling using a finite mixtures approach [29] with the MCLUST package in R to determine the optimal number of age subgroups for each chosen breed. We assumed the variable/unequal variance for the one-dimensional clustering and tested the number clusters between 1 and 4. The analysis determined the most suitable number groups by evaluating the Bayesian Information Criterion (BIC) and identifying the corresponding number of subgroups. The age distribution of each cluster was then examined to determine if they fit into the “delayed birth notification” or “phantom entries” categories. If these groups were considered plausible, they were excluded from the final weight calculation.

Our objective was to develop an algorithm that estimated a weight indicating the plausibility of a given age for each selected breed. Once the algorithm converged, individuals were grouped according to the probability of belonging to each cluster, considering the existing group membership probabilities.

Based on the age distribution of each cluster, we excluded clusters with lower age distributions from the final analysis in 20 selected breeds. This step aimed to reduce bias towards higher mortality rates in young adulthood, thereby improving the accuracy of estimating life expectancy in later years. Additionally, the clusters with the highest age distributions were excluded in 12 breeds to minimize uncertainty associated with implausibly old dogs (“phantom” entries).

Following the analysis of descriptive statistics to summarize the demographic characteristics of the sample, a hypothetical life table was generated for the Portuguese companion dog population, encompassing all dogs in the dataset. Dogs recognized by the UK Kennel Club (KC) [30] or Fédération Cynologique Internationale (FCI) [31] were classified as purebred, while all others were categorized as crossbreeds in the dataset. A cut-off of 100 dogs per breed was established for creating life tables and determining each breed’s life expectancy at birth. A total of 79 purebreds and 1 crossbreed met this criterion and were included in the analysis for differences in size and cephalic index. Data on body size were sourced from KC nomenclature, while the classification of the cephalic index, which describes the shape of their skull, was based on the FCI literature and previous studies [6]. This information was used to categorize the animals into three groups based on their body size (small, medium, or large) and cephalic index (brachycephalic, mesocephalic, or dolichocephalic).

Statistical analysis was performed using SPSS 28.0.1.0 and R version 4.3.2. A sub-analysis was performed on the estimated life expectancy without taking into account the variability within estimated values. A paired sample t-test was conducted to assess differences in life expectancy at birth between the sexes. To analyze the effect of body size and cephalic index on life expectancy at birth, general linear modeling was employed. Full factorial models were specified to account for potential interactions among independent variables. Effect sizes were calculated using partial eta-squared, which estimated the proportion of variance in the dependent variable explained by each independent variable. The total variance explained by the model was the sum of all partial eta-squared values. A *p*-value of less than 0.05 was considered statistically significant.

### 2.3. Construction of Life Tables

After summarizing the demographics of the cleaned data using descriptive statistics, a life table was constructed for the ten-year time frame using the Chiang adjusted method [15,32]. The construction of a life table involves the use of a one-year age interval (x, x + 1) and focuses on two primary variables—the number of individuals alive at age x, and dx, representing the number of deaths occurring within the same age interval (x, x + 1). The central death rate [m(x)] is calculated as the ratio of the number of dead dogs to the total study population, and the conditional probability of death in a one-year age interval [q(x)] is calculated as the product of the central death rate and the proportion of pets surviving to the end of the age interval [a(x)], under the assumption that deaths are distributed evenly across the entire year. To calculate the number of pets surviving to age x [l(x)], we assumed a hypothetical population of 100,000 pets at birth and applied an estimated mortality regime to simulate their survival over time. The number of pet years lived at age x [L(x)] was calculated as the sum of pets surviving until age (x + 1) and the fraction of pets surviving in age interval (x, x + 1) was simplified to [l(x) + l(x + 1)]/2.

The total number of years lived beyond age x was calculated by summing the number of years lived in each age interval starting from age x, T(x) = ∑i = xωL(x), where ω is the number of age intervals. Life expectancy at year x was then calculated as e(x) = T(x)/l(x), where l(x) is the number of pets surviving to age x. Confidence intervals (CIs) at the 95% level were calculated for each year using the mean e(x) and standard deviation. For each subpopulation, the variables P(x) and d(x) specific to that subpopulation were used to recalculate the intermediate variables, m(x), q(x), l(x), L(x), and T(x).

## 3. Results

A total of 310,559 dogs with confirmed deaths were selected from the SIAC records between 1 January 2013 and 31 December 2022. After applying the mixed-modeling approach as described in the methods, a refined dataset of 278,116 dogs remained, providing a reliable foundation for further analysis.

The final analytic dataset included 286 purebred breeds, with 194,146 (69.8%) purebred dogs, and 83,970 (30.2%) mixed-breed dogs. There were, in total, 138,325 females (49.7%) and 139,791 males (50.3%).

Table 1 provides an overview of the current lifespan of registered dogs of all breeds and crossbreed combined. The data revealed that the mean life expectancy at birth for a dog born in Portugal was around 8.91 years (8.43–9.39).

The female dogs tended to have a similar lifespan to the male dogs, with a mean lifespan at birth of 8.95 years (8.51–9.39), compared to 8.88 years (8.37–9.39) for the males, as shown in Figure 1 and Appendix A (Females) and Appendix A (Males). The trend in life expectancy across age groups appears to be similar regardless of sex.

### 3.1. Life Expectancy at Birth for Most Common Breeds (Non-Portuguese)

The findings presented in Table 2 summarize the results for the top 20 non-Portuguese breeds, each comprising a minimum of 1000 individuals and collectively representing 54% of the dataset. At age 0, the Yorkshire Terrier exhibited the highest life expectancy (11.7 years), while the French Bulldog showed the lowest life expectancy (6.29 years). Results of life expectancy at birth for both females and males across the top 20 breeds are also shown in Table 2.

For the majority of breeds (70%), females exhibited higher life expectancy at birth compared to males, although these differences were not statistically significant [t(9) = −1.310. *p* = 0.206)].

Life tables constructed for the top ten individual breeds (Portuguese and non-Portuguese) are available in the Appendix A.

Figure 2 enables a comparison of life trajectories among three breeds with shorter lifespans (indicated by dashed lines) and three breeds with longer lifespans (represented by a solid line). As dogs progress into their senior years, the differences in life expectancy become less evident.

### 3.2. Life Expectancy at Birth for Portuguese Dog Breeds

The results for the top 10 most frequent Portuguese breeds are outlined in Table 3. The mean life expectancy at birth for the Portuguese breeds was 8.3 years. The mean life expectancy at birth for the female Portuguese breeds was 8.34 years and for male Portuguese breeds was 8.15. More than half (60%) of the female Portuguese breeds presented a higher life expectancy at birth though these differences were not statistically significant. [t(9) = −1.409. *p* = 0.192)].

Among the ten Portuguese breeds, only one was classified as small, while half of them fell into the large size category, and the remaining four were categorized as medium-sized. At age 0, the Portuguese Water Dog displayed the highest life expectancy (10.85 years), while the Transmontano Mastiff exhibited the lowest life expectancy (6.71 years).

Figure 3 facilitates a comparison of the life paths of various Portuguese dog breeds. It specifically identifies the top three breeds with shorter lifespans (represented by dashed lines) and three breeds with longer lifespans (represented by solid lines).

### 3.3. Life Expectancy at Birth by Cephalic Index and Body Size

Among the 79 selected breeds, 22 were categorized as small (27.9%) with 101,251 individual dogs, 28 breeds (35.4%) were classified as medium with 34,630 dogs, while large-sized dogs constituted 29 breeds (36.7%) with 52,436 dogs. When examining the cephalic index distribution among the 79 breeds, 15 were identified as brachycephalic (18.8%) with 12,636 dogs, 16 as dolichocephalic (21.3%) with 101,064 dogs, and 48 as mesocephalic (60%) with 74,617 dogs.

An analysis of the data revealed that smaller breeds had the highest mean lifespan at birth, with a mean of 9.52 years, followed closely by medium-sized breeds with a mean of 9.26 years, and larger breeds with a mean of 8.53 years. When the breeds were grouped by head size, the data showed that the brachycephalic breeds had the lowest mean lifespan at birth with a mean of 7.99 years, compared to the dolichocephalic breeds with a mean of 8.86 years and the mesocephalic breeds with a mean of 9.47 years.

Further exploration within each cephalic index category revealed that large brachycephalic breeds, such as Neapolitan Mastiffs, Spanish Mastiffs, or Cane Corso, exhibited shorter a life expectancy at birth. Among small brachycephalic breeds, the French Bulldog stood out as an outlier with unusually short lifespan of 6.29 years, which was the lowest among all breeds.

In dolichocephalic breeds, while large breeds had the lower mean life expectancy, the gap between medium and large breeds was narrower with substantial diversity among individual breeds, Finally, among mesocephalic breeds, small breeds like the Yorkshire Terrier and Bichon Frisé presented the highest life expectancy at birth of the dataset.

The general linear model univariate analysis revealed that life expectancy at birth was significantly associated with body size and cephalic index. The effect size was considered large for cephalic index [F(2,76) = 6.93, *p* = 0.002, η^2^ = 0.165)] and medium for body size [F(2,76) = 5.25, *p* = 0.008, η^2^ = 0.13)]. The interaction between life expectancy, body size, and cephalic index was not significant (*p* > 0.05). These results are illustrated in Figure 4. 

## 4. Discussion

It is undeniable that proper care and attention to various aspects of a pet’s life can significantly impact their longevity and overall health. Providing proper care, including regular exercise and mental stimulation, are essential for extending a pet’s lifespan [33]. Moreover, a combination of high-quality nutrition, appropriate husbandry, and healthcare have been identified as important factors in achieving exceptional longevity [34,35]. Most of our understanding of canine longevity comes from analyzing various data sources, including veterinary medical center records [6,9,10,36], private practice medical records [4,16,37], pet insurance company data [2,38,39,40,41], owner surveys [5,8,12,13,42], and even pet cemetery records [25,26]. While each source provides valuable insights, they also have limitations that affect their reliability and representativeness.

A key strength of our study is its broad scope, which enabled us to analyze data from over 3 million dogs across the country. This research benefits from a second important strength, the data’s high representativity, derived from the use of a comprehensive national database encompassing information on all registered dogs in Portugal. This vast repository of information helps mitigate selection and representation biases, thereby enhancing the accuracy and completeness of the findings. Furthermore, the fact that microchipping has been compulsory since 2008 contributes to the dataset’s robustness, yielding a detailed portrait of the Portuguese dog population.

The general life expectancy at birth for the dogs included in this study was lower than the life expectancy of dogs from the United Kingdom (11.23 years) [27], United States of America (12.69 years) [3], and Japan (13.7 years) [26]. It is crucial to acknowledge that all these studies were conducted in different countries, at different points in time and with different sample populations. The lack of a standardized method for reporting and recording dog mortality data across these studies further complicates the assessment of lifespan accuracy. Variations in methodology, study groups, and factors such as weight and breed make it challenging to compare and derive definitive conclusions from the data available. Analyzing canine lifespan through mortality data often excludes individuals from the same birth year who were still alive at the time of data collection. This exclusion can result in lifespan underestimation, known as right-censoring, since the full lifespan of these individuals is not fully captured [17]. Techniques like the Kaplan–Meier analysis and Cox regression can mitigate this issue in longitudinal studies with multiple data points [16,22,37]. However, the retrospective nature of our study and the limitations of available data make it challenging to directly apply these methods and compare lifespan estimates across different studies.

Our study found the top seven breeds with an mean lifespan exceeding 11 years; five were small-sized (American Cocker Spaniel, Bichon Frisé, English Cocker Spaniel, West Highland White Terrier and Yorkshire Terrier) and two were medium-sized breeds (Standard Schnauzer and Samoyed). When analyzing the relationship between life expectancy at birth, cephalic index and body size, the findings became more evident and consistent with the prior research, indicating a link between smaller body size and increased lifespan in dogs [3,16,26,37].

On the other side, findings about the breeds with the lowest life expectancy are also similar to other studies [3,16,26], Among the breeds with a life expectancy at birth of just under 7 years old were six large breeds (Spanish Mastiff, Transmontano Mastiff, Neapolitan Mastiff, Great Dane and Cane Corso) and one small breed (French Bulldog). The shorter lifespan of large breed dogs may be attributed to several possible mechanisms. Artificial selection has played a significant role in shaping the characteristics of dog breeds, and it is plausible that the emphasis on size has inadvertently led to the preservation of genetic factors that influence accelerated aging, leading to a reduced lifespan [6,10]. Factors such as oxidative stress and cellular metabolism have been implicated in the differences in longevity between large and small breed dogs [42,43,44]. Telomere length, which is associated with aging and longevity, has also been linked to individual breed lifespan. Studies have shown that telomeres shorten with age in dogs at a rate that is proportionate to the difference in mean lifespan between dogs and humans. Additionally, the length of telomeres varies between breeds, with larger breeds such as the Great Dane having shorter telomeres [45].

Our study’s findings also indicate that among the 10 dog breeds with shorter lifespans, brachycephalic breeds account for half of the total. This is consistent with the previous research [27,46] which highlighted the reduced longevity of flat-faced breeds compared to their non-brachycephalic breeds.

French Bulldogs have the lowest life expectancy at birth among all breeds, a trend potentially influenced, to some extent, by their swift rise in popularity in recent years. According to data from our records in SIAC, the number of French Bulldogs registered in the Portugal rose from 1234 individuals in 2013 to 7554 in 2022. This surge in popularity means that there are proportionally more young animals in this population compared to other breeds, decreasing their life expectancy.

Purebred dogs exhibited both longer and shorter lifespan estimates compared to crossbred dogs. Approximately 42% of purebreds lived longer than crossbreds, challenging the common belief that crossbreds are universally healthier. Furthermore, this observation aligns with recent findings published [7].

The findings of our study suggest that both body size and cephalic index significantly influence the life expectancy of dogs at birth. The effect size of cephalic index was considered large, while that of body size was considered medium. This implies that cephalic index has a more substantial impact on life expectancy compared to body size. However, there was no interaction effect between body size and cephalic index on life expectancy, indicating that these two factors independently influence the life expectancy of dogs. The absence of an interaction effect suggests that the relationship between life expectancy and one factor (e.g., body size) remains consistent regardless of the level of the other factor (e.g., cephalic index). These results contribute to our understanding of the factors that influence the life expectancy of dogs, which could potentially aid in predicting life expectancy and developing strategies to improve the health and longevity of dogs.

For the specific analyses considering size and cephalic index, we focused exclusively on 79 purebred breeds, each with a minimum of 100 individuals. This approach was chosen to ensure statistical reliability and consistency in our data. Mixed-breed dogs were excluded from these analyses due to their significant variability in size and head shape. In future studies, gathering more detailed data on mixed-breed dogs, including precise measurements of their size and head shapes, could enable their meaningful inclusion in cephalic index and size analyses, providing a more comprehensive overview of life expectancy across the entire canine population.

Our study did not thoroughly investigate the influence of non-biological factors, including owner demographics, management styles, and breed function, on the likelihood of early mortality in the Portuguese canine population. Moreover, it did not examine the specific causes of death for various dog breeds across different age groups. This absence of data regarding the reasons for mortality may introduce bias, given that distinct breeds could exhibit diverse susceptibilities to specific diseases or health issues. Additionally, the practice of euthanasia might result in a shorter overall lifespan compared to the potential lifespan if the dogs were allowed to pass away naturally.

The accuracy of certain information provided by dog owners, such as breed and date of birth or death, could not be verified during data collection. For instance, the absence of cross-checking with pedigree records may have resulted in dogs of mixed breeds being mistakenly classified as purebred, potentially inflating the numbers of certain breeds, such as the Portuguese Podengo. Furthermore, the accuracy of the age at death information may have been compromised in some cases due to owners’ uncertainty regarding their dog’s precise date of birth, particularly when the dog had been adopted or rehomed. Additionally, the reported date of death could also be inaccurately documented, either because pet owners delayed notifying the practice or due to entry errors made by veterinary staff. Also, collecting data on whether dogs died naturally or were euthanized would help in understanding the factors contributing to mortality and life expectancy estimates.

To mitigate this bias, rigorous data cleaning methods were employed, enhancing the accuracy of the final dataset, although some residual uncertainty persists. We cross-referenced age at death distributions for consistency against databases provided in the other studies [4,8,12,27,47]. Although the reduced dataset may have affected the representation of the original study population, the initial datasets were sufficiently large to compensate for this.

Overall, understanding the factors that influence canine longevity can contribute to improving the health and wellbeing of companion animals. By examining the relationship between breed size and lifespan, researchers can identify potential strategies for promoting healthy aging in dogs.

## Figures and Tables

**Figure 1 animals-14-02141-f001:**
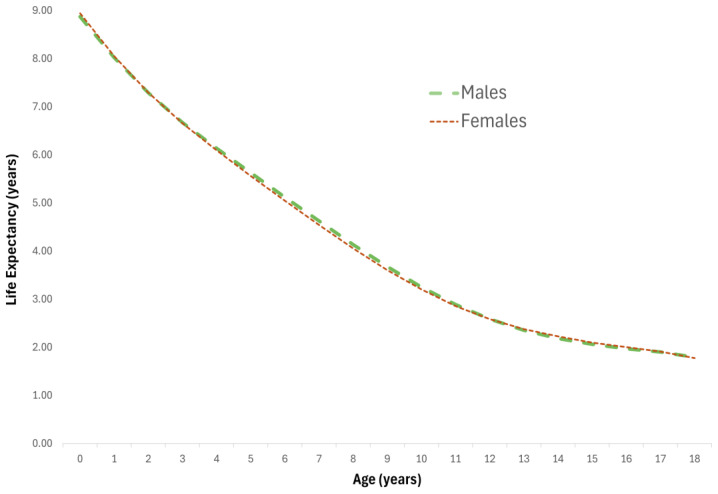
Life expectancy for female and male dogs across different ages.

**Figure 2 animals-14-02141-f002:**
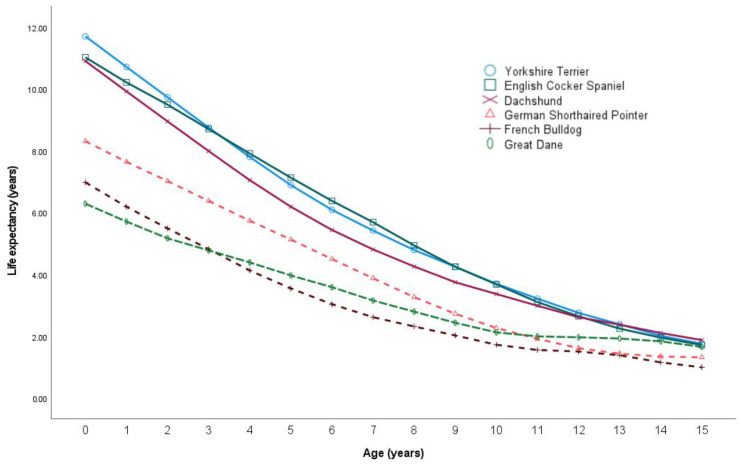
Life expectancy comparison between top three breeds with shorter lifespan (represented by dashed lines) and top three breeds with a longer lifespan (represented by a solid line) across different ages among the 20 most common non-Portuguese dog breeds.

**Figure 3 animals-14-02141-f003:**
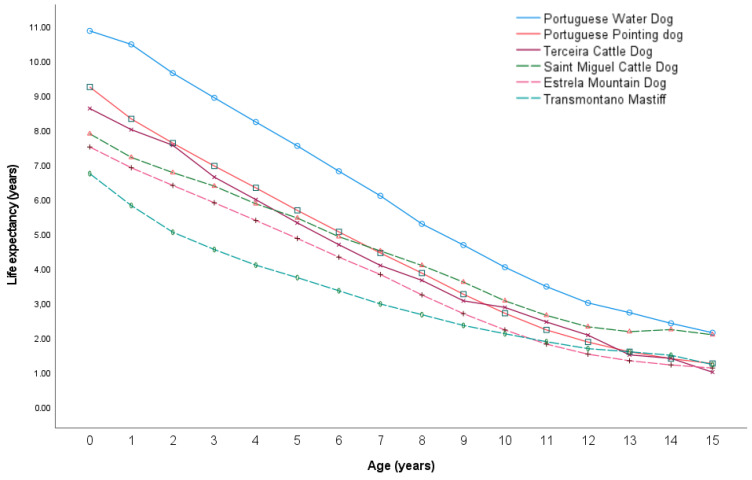
Life expectancy comparison between three Portuguese breeds with shorter lifespans (represented by dashed lines) and three Portuguese breeds with a longer lifespan (represented by a solid line) across different ages.

**Figure 4 animals-14-02141-f004:**
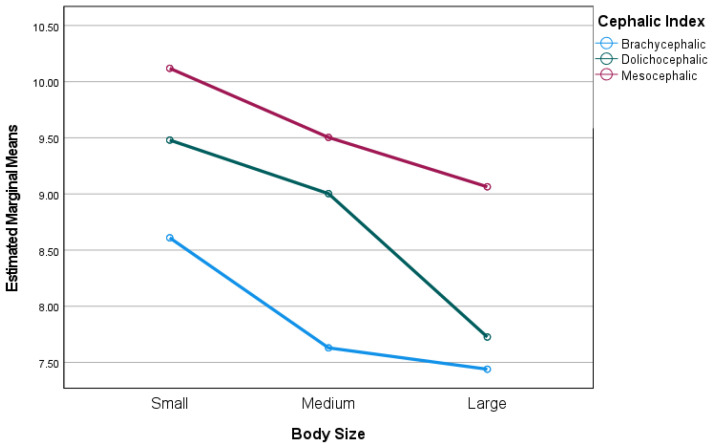
Plot showing marginal means of life expectancy by body size for each cephalic index.

**Table 1 animals-14-02141-t001:** Current life table of Portuguese companion dogs registered in SIAC (all pure breeds and crossbreed combined).

Age Interval in Years (x, x + 1)	Number Living at Age x l(x)	Number Dying in the Age Interval (x, x + 1) d(x)	Conditional Probability of Death q(x)	Proportion Surviving to Age x a(x)	Number of Dog-Years Lived at Age x L(x)	Total Number of Dog-Years Lived from Year x t(x)	Life Expectancy for Dogs in the Age Interval (x, x + 1) e(x)
0–1	278,116	4158	0.01	1.00	99,258	891,040	8.91 (8.43–9.39)
1–2	273,958	9145	0.03	0.99	96,899	791,782	8.04 (7.36–8.71)
2–3	264,813	14,143	0.05	0.95	92,803	694,883	7.29 (6.49–8.10)
3–4	250,670	17,632	0.07	0.90	87,256	602,080	6.67 (5.79–7.54)
4–5	233,038	19,150	0.08	0.84	80,865	514,824	6.12 (5.21–7.02)
5–6	213,888	19,529	0.09	0.78	74,157	433,959	5.60 (4.69–6.50)
6–7	194,359	20,130	0.10	0.71	67,287	359,802	5.08 (4.18–5.99)
7–8	174,229	20,490	0.11	0.64	60,259	292,515	4.58 (3.68–5.49)
8–9	153,739	21,644	0.13	0.57	52,986	232,256	4.10 (3.17–5.02)
9–10	132,095	22,591	0.16	0.49	45,376	179,270	3.64 (2.69–4.59)
10–11	109,504	23,262	0.19	0.41	37,512	133,893	3.23 (2.25–4.21)
11–12	86,242	22,518	0.23	0.34	29,656	96,382	2.87 (1.87–3.88)
12–13	63,724	20,011	0.27	0.26	22,285	66,726	2.59 (1.56–3.62)
13–14	43,713	16,041	0.31	0.19	15,874	44,441	2.37 (1.31–3.42)
14–15	27,672	11,297	0.34	0.13	10,764	28,567	2.20 (1.14–3.26)
15–16	16,375	7317	0.37	0.09	7002	17,803	2.08 (1.01–3.15)
16–17	9058	4267	0.38	0.05	4401	10,801	1.99 (0.92–3.05)
17–18	4791	2287	0.39	0.03	2716	6400	1.90 (0.87–2.93)
18–19	2504	1201	0.39	0.02	1668	3684	1.78 (0.81–2.76)
19–20	1303	680	0.41	0.01	1006	2016	1.59 (0.66–2.52)
20–21	623	382	0.47	0.01	569	1010	1.36 (0.48–2.24)
21–22	290	241	0.59	0.00	279	441	1.12 (0.23–2.01)
22–23	49	49	1.00	0.00	163	163	1.00 (0.02–1.98)

**Table 2 animals-14-02141-t002:** Key statistics extracted from life tables for 20 most common non-Portuguese dog breeds in decrescent order of life expectancy at birth.

Breed	Population Size	Life Expectancy at Age 0	Life Expectancy at Age 0 for Males	Life Expectancy at Age 0 for Females	Body Size	Cephalic Index
Yorkshire Terrier	3533	11.70 (11.55–11.86)	11.80 (11.64–11.96)	11.61 (11.46–11.75)	Small	Mesocephalic
English Cocker Spaniel	2044	11.02 (10.44–11.60)	10.91 (10.34–11.49)	11.13 (10.54–11.72)	Small	Mesocephalic
Dachshund	1281	10.98 (10.89–11.08)	11.01 (10.68–11.18)	10.89 (10.72–11.07)	Small	Dolichocephalic
Golden Retriever	2167	10.97 (10.45–11.49)	11.06 (10.56–11.56)	10.87 (10.31–11.42)	Large	Mesocephalic
Poodle	4116	10.85 (10.25–11.45)	10.74 (10.20–11.29)	10.96 (10.29–11.62)	Medium	Dolichocephalic
English Pointer	1493	10.78 (10.64–10.92)	10.77 (10.64–10.90)	10.80 (10.65–10.94)	Medium	Mesocephalic
English Setter	1516	10.41 (10.24–10.57)	10.36 (10.19–10.54)	10.45 (10.29–10.60)	Large	Mesocephalic
Brittany Spaniel	7903	10.28 (10.08–10.48)	10.44 (10.24–10.64)	10.13 (9.93–10.33)	Medium	Mesocephalic
Dalmatian	1019	10.22 (9.64–10.79)	9.97 (9.42–10.52)	10.56 (9.95–11.17)	Medium	Mesocephalic
Labrador Retriever	13,263	9.77 (9.03–10.50)	9.68 (8.94–10.42)	9.88 (9.16–10.61)	Large	Mesocephalic
Crossbreed	83,970	9.48 (9.13–9.84)	9.35 (8.99–9.70)	9.64 (9.28–10.00)	NA	NA
Rottweiler	3406	9.30 (8.75–9.85)	9.29 (8.78–9.81)	9.31 (8.73–9.90)	Large	Mesocephalic
Boxer	3432	9.15 (8.59–9.71)	9.10 (8.52–9.68)	9.22 (8.68–9.76)	Large	Brachycephalic
Beagle	4064	9.06 (8.89–9.23)	9.27 (9.08–9.46)	8.96 (8.81–9.10)	Small	Mesocephalic
German Shorthaired Pointer	4852	8.83 (8.38–9.29)	8.77 (8.33–9.27)	8.93 (8.61–9.25)	Medium	Mesocephalic
Miniature Pinscher	2921	8.74 (8.20–9.29)	8.64 (8.08–9.20)	8.88 (8.35–9.40)	Small	Dolichocephalic
Chihuahua	1446	8.46 (8.03–8.89)	8.57 (8.13–9.02)	8.40 (7.99–8.81)	Small	Brachycephalic
German Shepherd Dog	11,542	8.31 (7.55–9.08)	8.26 (7.53–8.99)	8.39 (7.58–9.20)	Large	Dolichocephalic
Great Dane	1186	6.98 (6.36–7.59)	6.90 (6.26–7.54)	7.05 (6.46–7.64)	Large	Dolichocephalic
French Bulldog	2726	6.29 (5.44–7.14)	6.22 (5.41–7.04)	6.32 (5.44–7.21)	Small	Brachycephalic

**Table 3 animals-14-02141-t003:** Key statistics extracted from life tables for 10 top Portuguese breeds in decrescent order of life expectancy at birth.

Breed	Population Size	Life Expectancy at Age 0	Life Expectancy at Age 0 for Males	Life Expectancy at Age 0 for Females	Body Size	Cephalic Index
Portuguese Water Dog	384	10.85 (9.80–11.89)	10.51 (9.47–11.55)	10.25 (9.17–11.33)	Medium	Mesocephalic
Portuguese Pointing dog	2560	9.23 (8.84–9.61)	9.35 (8.95–9.75)	9.10 (8.74–9.47)	Medium	Mesocephalic
Terceira Cattle Dog	104	8.62 (7.78–9.46)	8.16 (7.05–9.27)	8.92 (8.41–9.43)	Medium	Mesocephalic
Castro Laboreiro Dog	810	8.44 (7.71–9.16)	8.14 (7.46–8.82)	8.78 (7.99–9.58)	Large	Mesocephalic
Alentejo Mastiff	3486	8.03 (7.73–8.33)	7.96 (7.67–8.24)	8.14 (7.82–8.46)	Large	Mesocephalic
Portuguese Sheepdog	435	7.93 (7.35–8.51)	7.88 (7.24–8.51)	7.99 (7.47–8.50)	Medium	Mesocephalic
Portuguese Podengo	77,706	7.86 (7.51–8.22)	7.78 (7.39–8.16)	7.92 (7.58–8.26)	Small	Dolichocephalic
Saint Miguel Cattle Dog	1553	7.81 (7.10–8.51)	7.81 (7.05–8.58)	7.80 (7.09–8.51)	Large	Mesocephalic
Estrela Mountain Dog	3536	7.50 (6.66–8.33)	7.06 (6.23–7.88)	7.91 (7.06–8.75)	Large	Mesocephalic
Transmontano Mastiff	1009	6.73 (6.35–711)	6.80 (6.42–7.19)	6.55 (6.18–6.92)	Large	Mesocephalic

## Data Availability

The data presented in this study are available on request from the corresponding author.

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
