# Peer review of "Investigating the Life Expectancy at Birth of Companion Dogs in Portugal Using Official National Registry Data"

_animals, 2024, doi:10.3390/ani14152141_

Round 1

Reviewer 1 Report

Comments and Suggestions for Authors

Throughout: When using the term “average” it needs to be specified if this is median or mean.

Line 14: While true, this is not a relevant statement for a scientific paper

Line 37-41: While this sounds nice, it is all uncited personal opinion. I would recommend either citing these facts, reducing this introduction to a simpler statement, or making it clear this is author’s opinion.

Line 61-71: This degree of detail is more suitable for a discussion, where results of this study can be compared to results of previous.

Line 84: The previous paragraph discusses the types of life tables, but then this one mentions generic “life tables” without any further specification. Either the previous paragraph should be removed/moved, or this one should discuss the specifics of which life table type is being used

Line 89: Source

Line 94 and throughout: Be consistent with “life tables” vs “lifetables”

Line 103-112: This should be summarized to a single sentence with much less detail, and a reference to the law for those who wish to know more.

Line 114-119: These phantom entries are introduced, but there is no discussion of what is done with these entries, how they are avoided or addressed, etc. Either provide some context and information or remove this section

Line 129: Clarify what information was considered “essential”

Line 159: Clarify is this is 100 dogs total or per category

Line 160-161: Explain what cut-offs were used for each group

Line 210: Is this difference statistically significant?

Line 212: Is this trend statistically significant?

Line 226: Is there a significant difference between breeds?

Line 229: Capitalization

Line 233: What is the cut-off between longer and shorter lifespans

Line 254: Capitalization

Line 301-307: References are needed for all of this

Line 323: Theorize as to why this difference exists

Comments on the Quality of English Language

Adequate, only minor spelling and grammar changes needed

Reviewer 2 Report

Comments and Suggestions for Authors

As companion dogs play more and more important roles in human families and societies, including as potential sentinels of human health and aging, it is critical that we have an accurate understanding of their expected lifespan. As environmental and socio-cultural influences on health and aging in both humans and dogs vary dramatically from one region to the next, it is likewise important to have data that captures dog mortality in these different environments. The authors rightly recognize the importance of understanding average lifespan, and this paper, which aims to “provide a comprehensive picture of the life expectancy of dogs in Portugal,” represents an important addition to the literature. I’d like to thank the authors for tackling a challenging analytical problem in order to contribute to this sector of our understanding and appreciate the opportunity to review this manuscript.

  1. General concept comments –

  1. Introduction:

  • As there is currently no standardized method used for calculating estimated lifespan, it is prudent of the authors to describe the different methods that have historically been used. I might suggest that it would enhance the point to include more information on how these methods compare with one another, and how/why each is problematic. In this way we have more clear expectations leading into the analysis. 

  • At the end of the introduction (and elsewhere in the manuscript), it is unclear as to which breeds are being evaluated. Here, the authors mention the “20 most common breeds.” Is this Globally? In Europe? In Portugal? There is likewise some confusion in the way the manuscript is currently articulated as to whether we are looking at 79 breeds or 286. As it is currently written, these two different numbers seem to refer to the same analyses and provokes some confusion.

  1. Materials & Methods:

  • The authors do not discuss certainty regarding DOB or DOD data. Surely there would be two groups based on the records being drawn from (e.g., dogs bred and owned are likely to have more certain DOBs vs. those adopted from shelter). It may be interesting to calculate these two groups (certain vs. uncertain) separately to see if there are significant differences in outcome.

  • The authors need to include justification for the specified exclusion criteria. 

  • References to how/where the Chiang method has been previously used (especially compared to other methods) should be included.

  • I have not come across clustering analysis in the context of data cleaning specifically for survival analysis. Can the authors please find and include a reference for using this in this context (and with a note on how it applies here)?

  1. Results:

  • Given my understanding of previously calculated lifespan estimates in the literature, it is somewhat surprising to see such a low estimate (though of course there are many factors at play). It would be helpful to see numbers on percentages of each “category” of dog.

  • The cephalic index has a large effect size. Has this been analyzed/discussed more on this in terms of the general population? 

  1. Discussion:

  • While the authors mention potential cultural influences on lifespan, given the large difference in their estimate compared to other recent studies, I would encourage a more in-depth discussion of cultural norms, rates of euthanasia, where the records are originating (e.g. were a lot of dogs shelter dogs?), etc.

  • As we know the inconsistent use of various methodologies is problematic for these analyses, I would likewise encourage more discussion of how methods/results compare to other studies – e.g. why such huge discrepancy in numbers compared to others (is it methodological or cultural?).

  1. Specific comments/line edits (scientific content)

  • Line 105 – The authors specifically mention “dangerous dogs” – any analysis of differences in LE for dogs categorized as such? What percentage of population were they?

  • It is unclear how many purebred breeds vs. individuals of those breeds are included. The way the numbers are currently presented is confusing (e.g. line 158 says 79 purebreds while line 203 and abstract say 286).

  • Line 163 – Excluding mixed-breeds from cephalic index/size calculations is problematic if basing other stats on total population, no?

  • Line 203 –  This study bases calculations on a population with double the number of purebred dogs compared to mixed breed dogs, and it is not clear what percentage of these purebreds are considered doli/brachy // large (e.g., at increased likelihood for lower life expectancy) and so we don’t know how it is going to influence representation in the total population – e.g. if a larger % of doli/brachy are purebreds and live shorter lives, and there are more purebreds in total pop, then you’re going to skew low.

  • Sections 3.1 and 3.2 grammar/punctuation needs to be reviewed.

Comments on the Quality of English Language

Section 3 in particular needs review of grammar and punctuation.
